# Hypothalamic Norepinephrine Concentration and Heart Mass in Hypertensive ISIAH Rats Are Associated with a Genetic Locus on Chromosome 18

**DOI:** 10.3390/jpm11020067

**Published:** 2021-01-23

**Authors:** Olga E. Redina, Svetlana E. Smolenskaya, Yulia K. Polityko, Nikita I. Ershov, Michael A. Gilinsky, Arcady L. Markel

**Affiliations:** 1Federal Research Center Institute of Cytology and Genetics, Siberian Branch of the Russian Academy of Sciences, 10 Lavrentieva Ave., 630090 Novosibirsk, Russia; svsmol@ngs.ru (S.E.S.); polityko.yulia@gmail.com (Y.K.P.); nikotinmail@mail.ru (N.I.E.); markel@bionet.nsc.ru (A.L.M.); 2Scientific Research Institute of Physiology and Basic Medicine, 4 Timakova Street, 630117 Novosibirsk, Russia; m.a.gilinsky@physiol.ru; 3Department of Natural Sciences, Novosibirsk State University, 2 Pirogova Street, 630090 Novosibirsk, Russia

**Keywords:** norepinephrine concentration in the hypothalamus, heart mass, QTL analysis, SNPs, ISIAH hypertensive rat strain

## Abstract

The relationship between activation of the sympathetic nervous system and cardiac hypertrophy has long been known. However, the molecular genetic basis of this association is poorly understood. Given the known role of hypothalamic norepinephrine in the activation of the sympathetic nervous system, the aim of the work was to carry out genetic mapping using Quantitative Trait Loci (QTL) analysis and determine the loci associated both with an increase in the concentration of norepinephrine in the hypothalamus and with an increase in heart mass in Inherited Stress-Induced Arterial Hypertension (ISIAH) rats simulating the stress-sensitive form of arterial hypertension. The work describes a genetic locus on chromosome 18, in which there are genes that control the development of cardiac hypertrophy associated with an increase in the concentration of norepinephrine in the hypothalamus, i.e., genes involved in enhanced sympathetic myocardial stimulation. No association of this locus with the blood pressure was found. Taking into consideration previously obtained results, it was concluded that the contribution to the development of heart hypertrophy in the ISIAH rats is controlled by different genetic loci, one of which is associated with the concentration of norepinephrine in the hypothalamus (on chromosome 18) and the other is associated with high blood pressure (on chromosome 1). Nucleotide substitutions that may be involved in the formation or absence of association with blood pressure in different rat strains are discussed.

## 1. Introduction

Hypertension is known to contribute to the development of such often fatal complications as cerebral stroke and myocardial infarction. Myocardial hypertrophy associated with hypertension, although not so pronounced in its initial clinical manifestations, ultimately leads to the development of cardiosclerosis, dangerous arrhythmias and heart failure. The development of myocardial hypertrophy in arterial hypertension is produced mainly by an increased afterload on the left ventricle. However, a lot of information has accumulated suggesting that the cause of myocardial hypertrophy in hypertension is far from unambiguous.

The dissociation between the blood pressure (BP) level and myocardial mass increase has been described both in patients with hypertension and in experimental hypertensive animals [1]. Thus, it was shown that the mass of the left ventricle in spontaneously hypertensive rats (SHR) is increased, not only in adult rats in which the blood pressure level is high, but also in young rat pups without obvious signs of arterial hypertension, which indicated the genetic origin of cardiac hypertrophy. Antihypertensive therapy in adult SHR rats led to a decrease in blood pressure. At the same time, there was some involution of myocardial mass, but only if the antagonist of the sympathetic system, alpha-methyldopamine, was used as a hypotensive drug. The use of another antihypertensive drug (hydralazine) led to an even greater decrease in blood pressure, but there was no involution of myocardial hypertrophy. The authors concluded that SHR rat myocardial hypertrophy is caused not only by an increase in blood pressure, but also by hormonal stimulation by the sympathetic adrenal and renin–angiotensin systems [2]. Another type of hypertension with myocardial hypertrophy was developed in rats receiving a high fructose diet (a condition similar to the metabolic syndrome of humans). It turned out that in this case the mechanism of cardiac hypertrophy is caused not so much by the degree of increase in blood pressure, but by an increase in the function of the sympathetic adrenal system [3]. When studying the role of adrenergic stimulation on the growth of cardiomyocytes in cell culture, it was shown that stimulation of α1-adrenergic receptors induces the growth and hypertrophy of neonatal cardiomyocytes and enhances the expression of contractile protein genes. Moreover, this effect did not depend on the mechanical activity of cells [4]. All this indicates the presence of additional factors, along with enhanced blood pressure, responsible for the development of myocardial hypertrophy in arterial hypertension, and these additional factors are directed to stimulation of the sympathetic adrenal and other neuroendocrine systems.

Recently, more and more attention has been paid to the role of the central nervous system and its sympathetic part as a kind of trigger in the pathogenesis of hypertension [5]. Using modern methods of visualizing the functioning of the brain (Fmri—functional magnetic resonance imaging) and constructing a neuroconnectome it has been shown that nerve connections between a number of brain structures, including the hypothalamus and the brainstem regions, are involved in maintaining the sympathetic tone and the regulation of blood pressure [6]. Hypothalamus, receiving extensive efferent information and having wide effector connections, is considered as the central integrating link of the nervous system. It functions as a regulatory center in maintaining the body’s homeostasis, controlling endocrine, autonomic and somatic reactions. It is the hypothalamus that plays the role of coordinator of the neuroendocrine and sympathoadrenal systems’ functions [7].

One of the most important hypothalamic neurotransmitters is norepinephrine. The paraventricular nucleus of the hypothalamus is thought to be the primary recipient of stress-related information, in particular, from the norepinephrine-releasing Locus Ceruleus (LC). The hypothalamus–LC axis controls input and output information from and to the peripheral autonomic nervous system, from and to the hypothalamic–adrenocortical and adrenomedullary systems with numerous brain-to-gland positive and negative (feedback) loops [8]. The neurons of the paraventricular nuclei, in which the central stimulator of the pituitary–adrenal system (stress system)—corticotropin releasing hormone (CRH) is synthesized, also receives noradrenergic stimulation [9,10]. Postganglionic sympathetic neurons innervate the myocardium, and myocardial hypertrophy associated with increased sympathetic tone is realized through stimulation of both myocardial alpha and beta adrenoreceptors [11].

The concentration of norepinephrine in the hypothalamus is an important factor in the regulation of the activity of the sympathetic nervous system and blood pressure [12,13]. The relationship between activation of the sympathetic nervous system and cardiac hypertrophy is shown in experimental rat models. Adult rats with spontaneously developing hypertension (SHR) are characterized by heart hypertrophy [14] and an increased content of norepinephrine in the hypothalamus [15,16]. Sympathetic hyperactivity in rats with heart failure is associated with an increase in norepinephrine in the paraventricular nucleus of the hypothalamus at rest, and in acute sympathetic activation under conditions of restraint stress in rats with heart failure, an impairment of the central noradrenergic system’s response was observed [17].

Based on the available information, in Inherited Stress-Induced Arterial Hypertension (ISIAH) rats simulating the stress-sensitive form of arterial hypertension, one can also suggest a connection between sympathetic activation and pathological changes in the myocardium. In the medulla of the adrenal glands of the ISIAH rats, an increased amount of chromogranin A was shown [18], which is an indicator of the catecholamines synthesis activation [19]. In addition, an increased concentration of epinephrine in the adrenal glands has been shown [20], which indicates an enhanced function of the sympathoadrenal system in ISIAH rats. ISIAH rats are also characterized by increased reactivity of the sympathoadrenal and hypothalamic–pituitary–adrenal systems [20], as well as morphological changes in the peripheral target organs, including the heart. In early postnatal ontogenesis in ISIAH rats, specific morphometric changes were described, such as an increase in the thickness of the walls of the ventricles and interventricular septum, indicating the formation of structural remodeling of the ventricular myocardium [21], and in adult rats there is an increase in the absolute and relative mass of the heart and signs of left ventricular hypertrophy compared with normotensive control rats [22].

It is known that an increase in the activity of the sympathetic nervous system is associated with the development of pathophysiological processes in the cardiovascular system and the development of hypertension [23]. Taking into account that ISIAH rats have characteristic signs of increased sympathetic tone at rest and its increased reactivity under stress, including a sharp increase in norepinephrine and adrenaline in blood plasma, we can assume that cardiac hypertrophy in ISIAH rats is, at least in part, the result of sympathetic tension. This is also indicated by the high expression of mRNA of the *Adrb1* (beta 1-adrenergic receptor) gene in the myocardium of intact ISIAH rats [24]. The involvement of sympathetic activation both in the functioning of the heart and in the development of hypertension in ISIAH rats is also supported by the fact that intravenous administration of antisense oligonucleotides directed to mRNA of beta 1-adrenergic receptors resulted in a long-term decrease in the blood pressure in ISIAH rats [24].

One of the approaches to studying the genetic control of physiological and pathophysiological traits is Quantitative Trait Loci (QTL) analysis, which allows the determination of the genetic loci associated with phenotypic traits. QTL analysis is also an effective approach for determining common loci for several traits, where genes with pleiotropic effects on the traits under study can be located. Previously, using QTL analysis, we described the genetic loci associated with the level of blood pressure and the mass of the adrenal glands, kidneys, and heart [25,26], as well as the loci associated with the plasma corticosterone concentration in hypertensive ISIAH rats [27]. All the studied traits were associated with several loci suggesting polygenic control of the manifestation of the listed characters of ISIAH rats. In addition, it was previously shown using the linear regression method (ANOVA) that the trait “concentration of norepinephrine in the hypothalamus” can be associated with several genetic loci in ISIAH rats [28]. In addition, previously, using data from transcriptome (RNA-Seq) analysis of several organs/tissues of ISIAH and control (Wistar Albino Glaxo (WAG) rats, as well as the available sequencing data for the genomes of 42 other rat strains and substrains, ISIAH rat-specific single nucleotide polymorphisms (SNPs) have been described, and significant genetic distances between the genotype of ISIAH rats and the genotypes of all other rat strains and substrains were shown [29].

The aim of this work was to conduct a genome-wide analysis of the trait designated as “concentration of norepinephrine in the hypothalamus” using QTL analysis and to describe the complete list of genetic loci associated with this trait. In addition, the goal of this work was to identify the loci associated with both the concentration of norepinephrine in the hypothalamus and the increased myocardial mass in ISIAH rats, which may contain genes that lead to cardiac hypertrophy associated with increased sympathetic stimulation. In the QTL of interest, nucleotide substitutions specific to the ISIAH rat strain, which may possibly contribute to the phenotypes under consideration, were identified and discussed.

## 2. Results

### 2.1. Determination of Loci Associated with the Concentration of Norepinephrine in the Hypothalamus

Determination of loci associated with the concentration of norepinephrine in the hypothalamus was carried out using hypertensive ISIAH and normotensive control WAG rats. The concentration of norepinephrine in the hypothalamus of ISIAH rats is significantly higher than that of WAG rats (Table 1).

A QTL analysis of this trait was performed using F_2_(ISIAH × WAG) male rats at the age of 6 months. The trait “concentration of norepinephrine in the hypothalamus” was analyzed for the first time. Figure 1 shows the LOD (**logarithm of odds)** plots for chromosomes.

The genome scan revealed one significant and several suggestive loci associated with concentration of norepinephrine in the hypothalamus, indicating the polygenic control of the trait. The description of these QTL is given in Table 2.

The effects of alleles on the trait are shown in Table 3. In rats with two ISIAH alleles at the loci on chromosomes 18, 6 (in the region of the D6Rat143 marker) and X, the value of the trait “concentration of norepinephrine in the hypothalamus” was significantly increased in comparison with animals with both alleles of normotensive WAG rats in these QTL. At the loci, on chromosomes 4 and 16, the opposite effect was observed—a decrease in the value of the trait in the presence of two alleles of ISIAH rats. At two loci, a significant change in the concentration of norepinephrine in the hypothalamus was associated with a heterozygous genotype. At the locus on chromosome 6 (in the region of the marker D6Rat75), the value of the trait in animals with a heterozygous genotype decreased, and at the locus on chromosome 14 the value of the trait increased in heterozygotes. Accordingly, the obtained results show that the effects of alleles of hypertensive ISIAH rats at different loci have a specific modulating effect on the trait.

### 2.2. Determination of Loci Associated with both the Concentration of Norepinephrine in the Hypothalamus and Heart Mass

Using the QTL Cartographer program, which allows one to determine the covariance of two or more characters, the character covariance of the norepinephrine concentration in the hypothalamus with heart mass was found at the locus on chromosome 18 in the region of marker D18Mgh1 (Figure 2). The description of the locus associated with the mass of the heart on chromosome 18 is given in Table 4.

The presence of ISIAH alleles in QTL on chromosome 18 (in the region of marker D18Mgh1) is associated with an increase in both the concentration of norepinephrine in the hypothalamus and heart mass (Figure 3).

### 2.3. Nucleotide Substitutions (SNPs) Detected in ISIAH but not in WAG Rats

In the QTL associated with the concentration of norepinephrine in the hypothalamus and heart mass, 240 SNPs (Appendix A) were identified in the transcribed regions belonging to the sequences of 82 genes. None of the substitutions found at the locus were characterized as having a high impact effect, but most of them can have a modifying effect (Table 5).

Nonsynonymous substitutions were analyzed using the Sorting Intolerant From Tolerant (SIFT) algorithm (Table 6). In two genes, *Slc4a9* (solute carrier family 4, member 9) and *Tcof1* (treacle ribosome biogenesis factor 1), nonsynonymous substitutions were characterized in the SIFT program as Deleterious, i.e., these substitutions are likely to have an effect on the structure and/or function of the proteins encoded by these genes. Comparison with the sequencing data of the genomes of 42 rat strains and substrains showed that a nonsynonymous substitution in the mRNA sequence of the *Slc4a9* gene (c.269C>T; p.Ala90Val) occurs only in ISIAH rats (Table 6). Several substitutions were found in the *Tcof1* gene (Table 6 and Appendix A) and the substitution c.1556C>A, leading to the replacement of the amino acid p.Ala519Glu, which is characterized as a Deleterious one, occurs in 20 out of 42 rat strains (Appendix A). 

## 3. Discussion

The genome scan was carried out and QTL associated with the concentration of norepinephrine in the hypothalamus of ISIAH rats were identified. QTL analysis of this trait on other rat strains has not yet been performed. The obtained results unambiguously show that the manifestation of the trait “concentration of norepinephrine in the hypothalamus” in ISIAH rats is under polygenic control, like other earlier studied physiological traits (blood pressure, body weight and mass of the adrenal glands, kidneys, and heart, as well as plasma corticosterone concentration) [26,27]. We showed that the presence of ISIAH hypertensive rat alleles in some loci can lead to a decrease in the concentration of norepinephrine in the hypothalamus, and in other loci—to its increase. These data may be useful in selecting candidate genes for a specific modulating effect on the trait.

The most highly significant association of norepinephrine concentration in the hypothalamus was found with a genetic locus on chromosome 18. The significant extent of this QTL suggests that, in this region of chromosome 18, there are most likely several genes that may be involved in the genetic control of the norepinephrine concentration in the hypothalamus in ISIAH rats. In the central part of chromosome 18 in the region of marker D18Mgh1, this QTL overlaps with the locus previously described as associated with heart mass [26]. Despite the fact that the genetic control of heart mass in ISIAH rats was earlier associated with several genetic loci, only one of these was found to be associated with both heart mass and norepinephrine concentration in the hypothalamus (Figure 2). Accordingly, it is precisely in the central part of chromosome 18 in the region of the D18Mgh1 marker the closely linked genes controlling the manifestation of the studied traits, or genes with a pleiotropic effect on the traits “concentration of norepinephrine in the hypothalamus” and “heart mass” can be localized. Earlier, for this locus on chromosome 18, an association with the heart mass (Cardiac mass QTL 125, Cm125) was also found in the congenic rats SS.LEW-(D18Chm41-D18Rat92)/Ayd, in which the fragment of chromosome 18 (30.8—52.3 Mb) of Dahl salt-sensitive (DSS) rats, which are a model of the salt-sensitive hypertension, has been replaced by a fragment of the genome of normotensive Lewis (LEW) rats [30].

Increased sympathetic activity is considered a risk factor for cardiovascular problems and as a cause of the onset and maintenance of high blood pressure [23]. The central nervous system, in particular the nuclei of the hypothalamus, plays a vital role in the regulation of these processes [31]. The current work made it possible for the first time to determine the genetic loci associated with the concentration of norepinephrine in the hypothalamus and compare them with the loci previously associated with heart mass and blood pressure both in ISIAH rats and in other hypertensive rat strains. When studying SS hypertensive rat strains simulating salt-sensitive hypertension [30,32,33,34], and in SHR rats with spontaneously developing hypertension [35], this locus on chromosome 18 was also associated with the Blood pressure trait. We mapped the Blood pressure trait earlier in ISIAH rats [26]; however, no associations of the Blood pressure trait with the genetic locus on chromosome 18 discussed in this work was identified. This is not the first time that certain loci have been associated only with the weight parameters of target organs, but not with the level of blood pressure. It is believed that such results make it possible to identify the mechanisms that underlie the development of hypertrophy of target organs but independent of hypertension [30]. Accordingly, we assume that the genetic control of sympathetic activation, which affects the development of hypertension in ISIAH rats, differs from that in SHR rats and rats with salt-sensitive hypertension. The differences found in the genetic control of the traits in SS, SHR, and ISIAH rats are in good agreement with the fact that the genotype of ISIAH rats is significantly different from the genotypes of other hypertensive strains [29]. The fact that no association with blood pressure was found in ISIAH rats at this locus is in good agreement with the concept of the nature of hypertension in humans, when chronic activation of the sympathetic nervous system in hypertension has a diverse range of pathophysiological consequences independent of any increase in blood pressure [36].

It is known that various (hemodynamic and hormonal) factors can influence the functioning of the heart, which can cause left ventricular hypertrophy [37]. Our results are in good agreement with these ideas. Despite the fact that in ISIAH rats at the locus on chromosome 18, the value of heart mass is not associated with the level of blood pressure; such an association was found on other chromosomes. We previously showed that in the distal part of chromosome 1 there is a locus associated with both blood pressure and heart mass in ISIAH rats [26]. Moreover, at the locus common to these two traits on chromosome 1, the presence of ISIAH rat alleles was associated with an increase in the values of both characters. Our results allow us to conclude that in ISIAH rats the development of heart hypertrophy has a complex nature and is associated both with sympathetic activation, which is controlled by the locus on chromosome 18, and with increased blood pressure, which is controlled by the locus on chromosome 1. These data are in good agreement with the idea that the activation of the sympathetic nervous system plays an important role in the development and maintenance of arterial hypertension and development of cardiac hypertrophy and heart failure. In addition, our data emphasize that these processes can be controlled by different genetic loci. This is consistent with the opinion that pharmacological treatment of hypertension should be aimed not only at lowering blood pressure, but also at correcting the sympathetic nervous activity [38].

In ISIAH rats, numerous nucleotide substitutions were found at the discussed locus, which can have a modifying effect on the level of gene expression. As shown in a number of studies, nucleotide substitutions in the regulatory regions of mRNA [39,40,41,42] and in introns [43,44] can have a significant modifying effect on transcription and translation and lead to various pathologies. Accordingly, it can be assumed that some of the substitutions listed in Table 5 may be essential in the formation of phenotypes associated with the discussed locus.

However, it is believed that it is the single nucleotide polymorphisms that lead to nonsynonymous amino acid substitutions in protein molecules that can most significantly affect its function and have a significant impact on human health compared to SNPs in other regions of the genome [45]. Our analysis allowed us to identify two SNPs that lead to nonsynonymous amino acid substitutions and, according to the SIFT algorithm, can presumably lead to changes in the structure and/or function of proteins encoded by the *Slc4a9* and *Tcof1* genes.

*Slc4a9* (solute carrier family 4, member 9) is encoding transmembrane anion exchange protein involved in chloride/bicarbonate exchange. However, according to the National Center for Biotechnology Information (NCBI) Database, this gene is expressed mainly in the kidneys, and its expression in brain and in heart is very low [46]. Accordingly, on the one hand, it can be assumed that the substitution found in the *Slc4a9* gene sequence may not play a key role in the formation of the discussed phenotypes. However, on the other hand, it can be assumed that the expression level of this gene may increase with sympathetic activation or other conditions and affect the function of the heart, which, however, has not been studied so far.

*Tcof1* (treacle ribosome biogenesis factor 1) gene product is involved in ribosomal DNA gene transcription by interacting with upstream binding factor [47]. Mutations in the *TCOF1* gene cause Treacher Collins syndrome (TCS), which is an autosomal dominant disorder characterized by abnormalities of craniofacial development that arises during early embryogenesis. However, the nucleotide substitution c.1556C>A, detected in the genome of ISIAH rats, also occurs in 20 out of 42 analyzed rat strains and substrains (see Appendix A), the development of which corresponds to normal, which indicates that the discussed substitution in the mRNA sequence of the gene *Tcof1* is not associated with TCS. Interestingly, the *Tcof1* gene is considered the only function candidate for Blood pressure in QTL 319 also known as C18QTL3 (53.3–78.6 Mb), described in the study of genetic control of hypertension in salt-sensitive rats [48].

Could a missense mutation in the *Tcof1* gene sequence be the reason that no association with blood pressure was found at this locus in ISIAH rats? As we wrote above, this mutation was found in 20 out of 42 analyzed rat strains/substrains. These included both several rat strains used as a normotensive control (FHL/EurMcwi, WKY/N, WKY/Gla, WKY/NHsd), and several hypertensive strains (FHH/EurMcwi, SBH/Ygl, SHRSP/Gla, SHR/OlaIpcv, SHR/NCrlPrin, SHR/NHsd, SHR/OlaIpcvPrin). However, it should be noted that among the hypertensive strains with this substitution, there were no strains with salt-sensitive hypertension, in which arterial blood pressure-associated loci were described in the discussed genome locus, and rats with spontaneous hypertension (SHR/Mol), for which an association with blood pressure was found at the locus of chromosome 18 [35] were not included in SNPs analysis. For SHRSP rats on chromosome 18, the Blood pressure QTL2 locus was described in region 1–34.9 Mb [49], which does not include the *Tcof1* gene, and for FHH [50] and SBH rats [51], Blood pressure trait mapping showed associations with other chromosomes. Based on the foregoing, it can be concluded that the found substitution in the *Tcof1* gene deserves additional attention and subsequent study of its possible association with the manifestation of hypertensive status in rats of different strains. As shown earlier, the epistatic hierarchy may play an important role in the genetic regulation of blood pressure [52]. Perhaps for the manifestation of this polymorphism on the regulation of blood pressure, its effect must be combined with other genetic factors.

The results presented in this work add new information to the existing knowledge about the functional annotation of the genome, which is an important step in understanding the formation of external phenotype [53]. Undoubtedly, the QTL method has its limitations and is only the first stage in a long process of further experimental evidence of the relationship between the function of candidate genes found in loci and the manifestation of the phenotype. Nevertheless, this work was the first to analyze the QTL for an important trait—the concentration of norepinephrine in the hypothalamus, and for the first time, a genetic locus associated with both the concentration of norepinephrine in the hypothalamus and an increased heart mass was described. In addition, it was shown that the locus found on chromosome 18 may or may not be associated with the level of blood pressure in different strains of hypertensive rats. Accordingly, the results obtained in this study can be useful for identifying specific targets for the treatment of heart pathology associated with increased sympathetic activity at the identified genetic locus on chromosome 18. Nucleotide substitutions specific to ISIAH rats that were identified and discussed in this study can potentially be used in further studies of genetic control of the manifestation of the corresponding phenotypes in human populations.

## 4. Materials and Methods

### 4.1. Animals

We used hypertensive ISIAH/Icgn rats (abbreviation from the words Inherited Stress-Induced Arterial Hypertension) and the normotensive WAG/GSto-Icgn (Wistar Albino Glaxo) strain. This work was carried out on the basis of the *Center for Genetic Resources* of *Laboratory Animals*, Institute of Cytology and Genetics, Siberian Branch of the Russian Academy of Sciences, Novosibirsk, Russia. Rats were kept under standard conditions, water and balanced food were given without restriction. All manipulations with animals were carried out in accordance with the European Convention for the protection of Vertebrate Animals Used for experimental and Other Scientific Purposes (ETS 123), Strasbourg, 18 March 1986 and in compliance with the rules of the Animal Care and Use Committee of Institute of Cytology and Genetics SB RAS, and approved by the Bioethical Committee of the Scientific Research Institute of Physiology and Basic Medicine (protocol No. 7 of 9 October 2015), Novosibirsk, Russia.

### 4.2. Determination of Norepinephrine Concentration in the Hypothalamus

Rats were quickly decapitated, the hypothalamus was isolated and frozen in liquid nitrogen, and then stored at −70 °C until the norepinephrine concentration was measured. A tissue sample was homogenized in a glass homogenizer in 300 μL 0.1 M perchloric acid, to which isopropyl–norepinephrine (100 ng/mL) (Sigma, United States) was added as an internal standard. The homogenate was centrifuged for 15 min (12,000× *g*). The supernatant was transferred to a separate tube. High performance liquid chromatography with two-electrode electrochemical detection was used for the analysis of norepinephrine in the supernatant [54]. The sensitivity of the measurement was 2 pg/mL. Norepinephrine concentration was measured in ng/mg of tissue.

### 4.3. QTL (Quantitative Trait Locus) Analysis

QTL analysis was performed on male F2 hybrids (ISIAHxWAG) at the age of 6 months (*n* = 126) using 149 polymorphic microsatellite markers. Their list and primer sequences are given on the website of the Institute of Cytology and Genetics SB RAS (http://icg.nsc.ru/isiah/en/category/qtl/). The position of microsatellite markers on chromosomes was determined using the RGSC Genome Assembly v 5.0. and expressed in millions of nucleotides (megabases, Mb) from the start of the chromosome. The genotyping was performed as described previously [26].

Linkage analysis was performed using the programs MAPMAKER/EXP 3.0 and MAPMAKER/QTL 1.1 [55]. The concentration of norepinephrine in the hypothalamus was transformed using the natural logarithm (ln) to reduce the asymmetry and excess (skewness and kurtosis) in the distribution of the values of the trait. A one-LOD drop-off was used to obtain an approximate 95% confidence interval for QTL position. QTL Cartographer Version 1.17, JZmapqtl [56] was used to map loci that are common for pairs of traits (bivariate analysis), and to calculate the LOD score significance thresholds. The level of statistical significance was calculated by random permutation of experimental data with replication 1000 times (permutation test) [57]. The linkage was considered significant if the experimentally obtained LOD score exceeded 5% of the threshold value in the analysis of the genome (experiment-wise threshold) [58], linkage was considered probable if the experimentally obtained LOD score exceeded 5% of the threshold value during a permutation test for a single chromosome (chromosome-wise threshold). Statistical processing of the results was carried out using Statistica 6.0 (StatSoft, Tulsa, OK, USA). The significance of differences between the mean values was evaluated using Student’s *t*-test. The degree of dominance was calculated by the standard method [59], assuming that the complete dominance of ISIAH rat alleles is +1, and the complete dominance of WAG rat alleles is −1.

The Rat Genome Database (RGD, http://rgd.mcw.edu/) was used to compare the genetic loci found in this work with the results of studies on other model animals.

### 4.4. Tissue Collection for SNP Analysis

To analyze SNPs, male ISIAH/Icgn and WAG/GSto-Icgn rats were used. Each experimental group consisted of 6 rats. Samples of five tissues were taken from each animal for analysis of the transcriptome (brainstem, hypothalamus, adrenal gland, cortex and medulla of the kidney). Sequencing of transcriptomes was performed at JSC Genoanalytica (Moscow, Russia).

### 4.5. RNA Sequencing

Over 10 million single-end reads of 50-bp length were obtained for each sample in accordance with standard Illumina protocols as described previously [29]. All samples were analyzed as biological replicates. 

Mapping was performed for the reference genome (Rnor_5.0\rn5) using the TopHat2 software [60]. The quality of the mapped data was assessed using the “CollectRnaSeqMetrics” module in the Picard software package (http://broadinstitute.github.io/picard/). Potential PCR duplicates were removed from the mapped bam data obtained for 5 different tissues of each animal (hypothalamus, brainstem, adrenal gland, renal medulla and cortex), after which they were combined into one pool for each animal with the Picard “MergeSamFiles” module for further analysis.

### 4.6. Polymorphism Detection

The initial set of polymorphisms was determined using the Genome Analysis Toolkit (GATK) [61] using the “HaplotypeCaller” module in the “GVCF” mode and the “GenotypeGVCFs” module for combined genotyping, using the settings for variant calling and hard filtering recommended by the GATK developers. The details were described in [29].

Determination of polymorphisms in transcriptome data of ISIAH and WAG rats was carried out relative to the reference genome sequence of BN/NhsdMcwi rats in the assembly version RGSC5.0 [62]. Further analysis of the list of polymorphic variants of ISIAH rats was carried out using the Rat Genome Sequencing Consortium data for the genome sequences of 42 rat strains and substrains including 11 hypertesive rat strains and substrains: FHH/EurMcwi, LH/MavRrrc, MHS/Gib, SBH/Ygl, SHR/OlaIpcv, SHRSP/Gla, SHR/NCrlPrin, SHR/NHsd, SHR/OlaIpcvPrin, SS/Jr, SS/JrHsdMcwi; 10 rat strains and substrains that usually serve as a normotensive control: FHL/EurMcwi, LN/MavRrrc, LL/MavRrrc, MNS/Gib, SBN/Ygl, SR/Jr, WKY/N, WKY/Gla, WKY/NCrl, WKY/NHsd; and 21 other rat strains and substrains that are used in experiments not related to hypertension: ACI/N, ACI/EurMcwi, BBDP/Wor, BN-Lx/Cub, BN-Lx/CubPrin, BN/SsN, BUF/N, DA/BklArbNsi, F334/N, F344/NHsd, F344/NCrl, SUO_F344, GK/Ox, LE/Stm (SOLiD), LEW/Crl, LEW/NCrlBR, LE/Stm (Illumina), M520/N, MR/N, WAG/Rij, WN/N [63]. Comparison of single nucleotide polymorphisms of ISIAH/Icgn and WAG/Gsto-Icgn strains with genotypes of 42 rat strains and substrains was carried out only for genomic loci sequenced in transcriptomic analysis.

### 4.7. Prediction of the SNP Effects

The classification of the found polymorphisms and their effects are given according to the description in the SnpEff program (http://snpeff.sourceforge.net/SnpEff_manual.html). To determine the possible effect of amino acid substitution on protein function, the Sorting Intolerant From Tolerant (SIFT) program [64] was used.

## Figures and Tables

**Figure 1 jpm-11-00067-f001:**
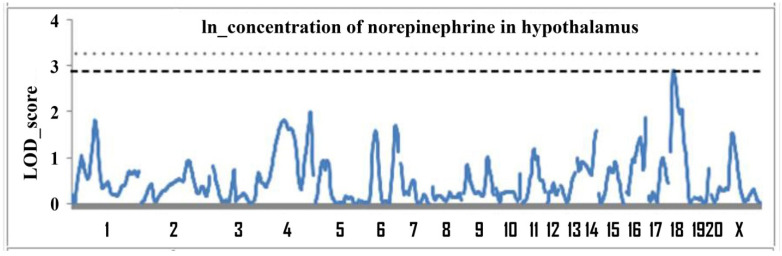
Genome-wide LOD (**logarithm of odds)** plots for concentration of norepinephrine in the hypothalamus in male rats F_2_(ISIAH × WAG) aged 6 months. The X axis shows the chromosome numbers. Lines show the level of experimentwise threshold: the dashed line corresponds to *p* = 0.05, the dotted line corresponds to *p* = 0.025.

**Figure 2 jpm-11-00067-f002:**
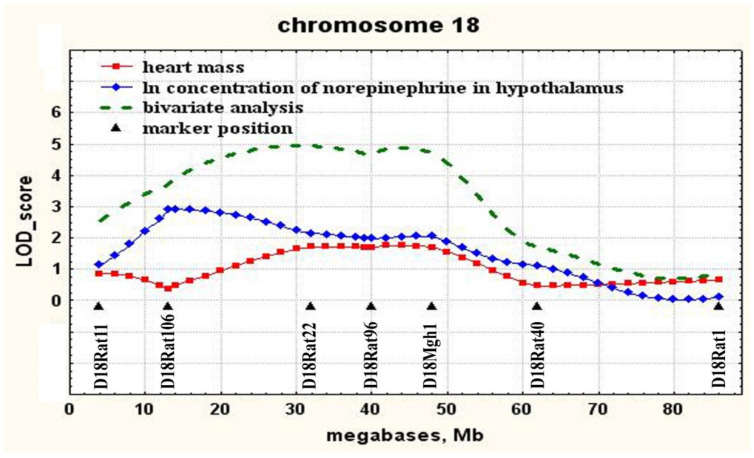
The covariance of two traits (the concentration of norepinephrine in the hypothalamus and heart mass) at the locus on chromosome 18.

**Figure 3 jpm-11-00067-f003:**
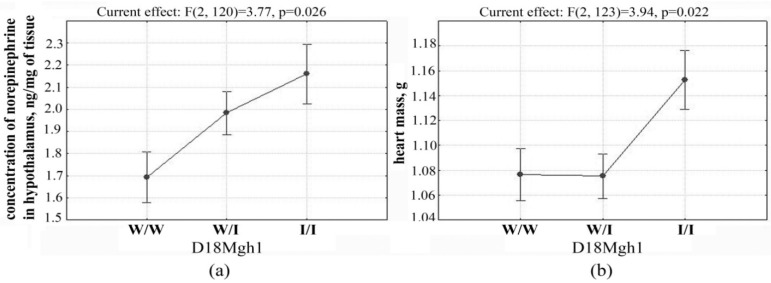
Allele effects on trait values in the QTL on chromosome 18 in the region of marker D18Mgh1: (**a**) concentration of norepinephrine in the hypothalamus; (**b**) heart mass; I/I—a homozygote for the ISIAH allele; W/W—a homozygote for the WAG allele; I/W—a heterozygote.

**Table 1 jpm-11-00067-t001:** Comparative characteristics of Inherited Stress-Induced Arterial Hypertension (ISIAH) and Wistar Albino Glaxo (WAG) rats aged 6 months.

Trait	ISIAH M ± SEM (*n* = 10)	WAG M ± SEM (*n* = 14)
Concentration of norepinephrine in the hypothalamus, ng/mg of tissue	1.698 ± 0.068 **	1.045 ± 0.068

** *p* < 0.01 in ISIAH as compared with WAG (Student’s *t*-test). M—mean, SEM—standard error of mean.

**Table 2 jpm-11-00067-t002:** Quantitative Trait Loci (QTL) associated with concentration of norepinephrine in the hypothalamus in male rats F_2_(ISIAHxWAG) aged 6 months.

Chr.	Peak Marker (Mb)	Confidence Interval *, Mb	LOD Score	*p*	Variability, %
**Significant QTL**					
18	D18Rat106 (13.2)	0–50	2.91	0.05	10.8
**Suggestive QTL**					
1	D1Rat30 (107.1)	90–128	1.82	0.05	6.6
4	D4Rat27 (137.8)	84–186	1.80	0.05	8.0
4	D4Rat68 (233.3)	214–242	1.99	0.025	7.2
6	D6Rat143 (48.1)	40–72	1.60	0.05	10.7
6	D6Rat75 (136.6)	120–156.9	1.71	0.05	7.8
14	D14Rat18 (86.0)	68–115.1	1.58	0.025	6.3
16	D16Rat48 (79.7)	72–90	1.87	0.025	7.3
X	DXRat26 (45.1)	28–70	1.54	0.05	6.1

* Boundaries of a locus are defined in the respective one LOD interval; Mb—megabases.

**Table 3 jpm-11-00067-t003:** Allele effects in QTL associated with concentration of norepinephrine in the hypothalamus in male rats F_2_(ISIAHxWAG) aged 6 months.

Trait Measurement in F_2_ Hybrids (ISIAH × WAG) n	Chr.	Peak Marker (Mb)	Genotype			D ^#^
			I/I M ± SEM *n*	I/W M ± SEM *n*	W/W M ± SEM *n*	
1.93 ± 0.07 123	**Significant QTL**					
	18	D18Rat106 (13.2)	2.21 ± 0.15 ** 29	1.97 ± 0.10 56	1.65 ± 0.11 38	0.1
	**Suggestive QTL**					
	1	D1Rat30 (107.1)	1.63 ± 0.12 ^††^ 27	2.05 ± 0.08 71	1.94 ± 0.18 24	−1.7
	4	D4Rat27 (137.8)	1.80 ± 0.11 * 31	1.79 ± 0.08 56	2.27 ± 0.16 ^††^ 36	1.0
	4	D4Rat68 (233.3)	1.87 ± 0.13 * 25	1.80 ± 0.07 74	2.39 ± 0.21 ^††^ 24	1.3
	6	D6Rat143 (48.1)	1.96 ± 0.11 * 50	2.01 ± 0.10 58	1.54 ± 0.13 ^††^ 15	1.2
	6	D6Rat75 (136.6)	2.11 ± 0.14 ^†^ 31	1.75 ± 0.08 59	2.09 ± 0.14 ^†^ 33	−35.0
	14	D14Rat18 (86.0)	1.72 ± 0.08 ^††^ 30	2.14 ± 0.12 58	1.76 ± 0.09 ^†^ 35	−20.0
	16	D16Rat48 (79.7)	1.61 ± 0.09 **^††^ 31	2.03 ± 0.10 60	2.06 ± 0.13 32	−0.9
	X	DXRat26 (45.1)	2.17 ± 0.13 ** 52		1.75 ± 0.06 71	

I/I—a homozygote for the ISIAH allele; W/W—a homozygote for the WAG allele; I/W—a heterozygote; Mb—megabases; * *p* < 0.05, ** *p* < 0.01—compared with W/W; ^†^
*p* < 0.05, ^††^
*p* < 0.01—compared with H/W; ^#^—degree of dominance.

**Table 4 jpm-11-00067-t004:** QTL on chromosome 18 associated with heart mass in male rats F_2_(ISIAHxWAG) aged 6 months.

Chr.	Peak Marker (Mb)	Confidence Interval *, Mb	LOD Score	*p* Chromosome-Wise	Variability, %
Heart Mass					
18	D18Mgh1 (47.7)	18–60	1.76	0.025	6.5

* Boundaries of a locus are defined in the respective one LOD interval; Mb—megabases.

**Table 5 jpm-11-00067-t005:** Classification of nucleotide substitutions (SNPs) effects.

SnpEff Classification	Number of SNPs	Effect
3_prime_UTR_variant	55	modifier
5_prime_UTR_premature_start_codon_gain_variant	3	low
5_prime_UTR_variant	3	modifier
downstream_gene_variant	67	modifier
intergenic_region	10	modifier
intron_variant	25	modifier
missense_variant	13	moderate
synonymous_variant	58	low
upstream_gene_variant	6	modifier

**Table 6 jpm-11-00067-t006:** Predicting the effects of missense variants on protein function using the Sorting Intolerant From Tolerant (SIFT) algorithm.

Gene Symbol	Position	ID	DP	SNP	Amino Acid Substitution	SIFT Score *	SIFT Classification
*Slc4a9*	29059899	novel	118	c.269C>T	p.Ala90Val	0.021	deleterious
*Dcp2*	35336920	novel	76	c.1109C>T	p.Ala370Val	0.718	tolerated
*Megf10*	51550880	rs199133377	131	c.3404C>G	p.Thr1135Ser	0.823	tolerated
*RGD1312005*	53356293	rs8169475	634	c.361A>G	p.Asn121Asp	1.000	tolerated
*Synpo*	55104621	rs63909326	331	c.1727T>C	p.Ile576Thr	0.378	tolerated
*Synpo*	55104676	rs198024246	253	c.1672C>T	p.His558Tyr	1.000	tolerated
*Tcof1*	55324879	rs198780519	351	c.2828A>G	p.Asn943Ser	1.000	tolerated
*Tcof1*	55334042	rs197009609	274	c.1556C>A	p.Ala519Glu	0.014	deleterious
*Tcof1*	55347410	rs199120971	231	c.126T>G	p.His42Gln	1.000	tolerated
*Csf1r*	55682600	rs198399348	798	c.1825C>A	p.Leu609Met	0.486	tolerated
*Hmgxb3*	55690668	rs198759766	350	c.3513T>G	p.His1171Gln	0.265	tolerated
*Hmgxb3*	55715142	rs197391293	293	c.1337G>A	p.Gly446Asp	0.087	tolerated
*Napg*	57562399	rs198931177	570	c.448T>A	p.Cys150Ser	0.157	tolerated

*—statistically significant values are given in bold; ID—identification number; DP—sequencing depth.

## Data Availability

Not applicable.

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
