# Peer review of "Hypothalamic Norepinephrine Concentration and Heart Mass in Hypertensive ISIAH Rats Are Associated with a Genetic Locus on Chromosome 18"

_jpm, 2021, doi:10.3390/jpm11020067_

Round 1
Reviewer 1 Report
The article provides original data on the genetic control of the concentration of norepinephrine in the hypothalamus of ISIAH rats that simulate a stress-sensitive form of arterial hypertension. This feature was mapped for the first time and is of undoubted interest. It is also interesting that the manuscript describes a locus in which the presence of alleles in ISIAH rats is associated with an increase in the concentration of norepinephrine in the hypothalamus and with an increase in the heart mass, but unlike some other lines of hypertensive rats, it is not associated with blood pressure. The results obtained are potentially interesting for studies aimed at studying the genetic control of these traits in humans. The identification of nucleotide substitutions in the discussed locus was carried out correctly, using a large array of genome sequencing data for other rat strains. Nucleotide substitutions have been identified that can potentially be used in further studies, including in human populations. In general, the manuscript is written in a clear manner and requires only minor editorial revisions. Notes are given in the text of the attached file.

Reviewer 2 Report
I enjoyed reading the article "Hypothalamic norepinephrine concentration
and heart mass in hypertensive ISIAH rats are associated with a genetic
locus on chromosome 18". The summary is consistent with the content of
the work. The reviewed work presents the current state of knowledge in a
logical, understandable, comprehensive, carefully and correctly citing
literature - 64 items. It does not contain factual errors, the names are
in accordance with generally accepted principles. In my opinion, the
text in English does not contain language errors.
Overall assessment of work very good.
My remarks focus on the order in which the content of the manuscript is presented. In my opinion the materials and methods (item 4) should be presented before the results and discussion (items 2 and 3). This will make this work clearer and more understandable.
Reviewer 3 Report
Redina and co-workers described a QTL analysis of ISIAH rats to determine the loci associated with increased noradrenaline and heart mass. The authors observed a positive correlation on chromosome 18 and they identified 82 genes with nucleotide substitutions. However, most of these mutations have no impact effect on the gene. In addition, the authors observed several nonsynonymous substitutions, including the solute carrier Slc4a9 and the ribosome biogenesis factor Tcof1.
The introduction is well written and very gives a very detailed overview of the current status of animal models with blood pressure. However, the result par is very short and several abbrevations are not explained which makes it difficult to read for non-experts. Some more details how the authors performed the test would be helpful. Why have the authors listed QTLs in table 2 which were not significant (below LOD of 3.3)? Since only the QTLs on chromosome 18 showed a significant effect it might be helpful to substantiate a potential regulation in gene and/or protein expression of the associated gene products in control, ISIAH rats and other disease models. Unfortunately, without any follow up experiments the suggested QTLs are highly speculative and it is unclear whether the discussed genes are associated with the disease model.
What means HNCAT in table 1?
